# Thermomechanical Processing for Improved Mechanical Properties of HT9 Steels

**DOI:** 10.3390/ma17153803

**Published:** 2024-08-01

**Authors:** Thak Sang Byun, David A. Collins, Timothy G. Lach, Jung Pyung Choi, Stuart A. Maloy

**Affiliations:** 1Oak Ridge National Laboratory, Oak Ridge, TN 37831, USA; collinsda@ornl.gov (D.A.C.); lachtg@ornl.gov (T.G.L.); 2Pacific Northwest National Laboratory, Richland, WA 99352, USA; jungpyung.choi@pnnl.gov (J.P.C.); stuart.maloy@pnnl.gov (S.A.M.)

**Keywords:** ferritic-martensitic steels, thermomechanical processing, carbide precipitates, strength and toughness

## Abstract

Thermomechanical processing (TMP) of ferritic–martensitic (FM) steels, such as HT9 (Fe–12Cr–1MoWV) steels, involves normalizing, quenching, and tempering to create a microstructure of fine ferritic/martensitic laths with carbide precipitates. HT9 steels are used in fast reactor core components due to their high-temperature strength and resistance to irradiation damage. However, traditional TMP methods for these steels often result in performance limitations under irradiation, including embrittlement at low temperatures (<~430 °C), insufficient strength and toughness at higher temperatures (>500 °C), and void swelling after high-dose irradiation (>200 dpa). This research aimed to enhance both fracture toughness and strength at high temperatures by creating a quenched and tempered martensitic structure with ultrafine laths and precipitates through rapid quenching and unconventional tempering. Mechanical testing revealed significant variations in strength and fracture toughness depending on the processing route, particularly the tempering conditions. Tailored TMP approaches, combining rapid quenching with limited tempering, elevated strength to levels comparable to nano-oxide strengthened ferritic alloys while preserving fracture toughness. For optimal properties in high-Cr steels for future reactor applications, this study recommends a modified tempering treatment, i.e., post-quench annealing at 500 °C or 600 °C for 1 h, possibly followed by a brief tempering at a slightly higher temperature.

## 1. Introduction

Core structures in advanced fast reactors are exposed to a wide temperature range of 300–550 °C and high dose (>200 dpa) irradiation at a high fast neutron flux (up to 10^16^/s·cm^2^) [1]. Over the past decades, considerable efforts have focused on developing fast reactor technologies for improved thermal efficiency and higher fuel burnup, such as achieving 40% burnup (~400 dpa in structural materials). Operating reactors at such high doses requires significant improvements in fuel cladding and duct materials [2,3,4,5,6,7], potentially necessitating the development of new reactor core materials [2]. Specifically, core materials for advanced fast reactors must possess high low-temperature toughness to prevent embrittlement due to neutron irradiation at lower temperatures (<~430 °C), as well as excellent resistance to creep and swelling at higher temperatures [2,3,4,6,7,8,9,10,11,12,13,14,15,16].

Compared to austenitic stainless steels, quenched and tempered ferritic–martensitic (FM) steels offer distinct advantages in void swelling resistance and thermal conductivity [2,3,4,7,8,9]. Consequently, the 12Cr–1MoVW (HT9) steels with tempered martensitic structures have been favored as primary core materials for fast reactors. They exhibit high resistance to irradiation-induced embrittlement, thermal and irradiation creep, and void swelling, and are compatible with liquid sodium coolant [2,3,4,5,6,7,8,9,10,11,12,13,14,15,16,17,18]. However, challenges remain, such as irradiation-induced embrittlement at low temperatures (<~430 °C), void swelling at very high doses (>200 dpa), and maintaining high-temperature strength above 500 °C, which limit the capabilities of HT9 steel components [2].

Meanwhile, the 9Cr FM steels are of particular interest due to smaller shifts in radiation-induced ductile–brittle transition temperatures compared to higher chromium content FM steels [13,19]. Although not yet used as core materials in operational reactors, 9Cr steels with lower chromium equivalents generally offer higher hardenability during quenching, enabling finer microstructure production compared to their 12Cr counterparts. This suggests potential advantages in achieving finer microstructures through improved thermomechanical processing (TMP) routes tailored for thin HT9 steel core components.

FM steels are typically quenched and tempered (Q and T) to achieve high strength and toughness by forming fine martensitic and ferritic laths and M_23_C_6_/MC carbide precipitates [20,21,22,23]. However, traditional normalization–quenching–tempering treatments often result in limited mechanical properties at high temperatures due to significant strength reduction above ~450 °C. This is attributed to high-temperature tempering, which induces coarsening of lath structures and carbides, and dislocation annihilation [20,21,22]. While high-temperature tempering maximizes ductility, it can compromise high-temperature strength and potentially reduce radiation resistance compared to steels with finer microstructures. Therefore, optimizing TMP routes for high-Cr steels could enhance their performance during high-temperature irradiation by refining microstructural features to directly engage radiation damage processes, such as forming nanoscale defect clusters, voids, helium bubbles, and element segregation [22,23,24,25,26,27,28,29].

Past efforts in optimizing TMP for nuclear reactor applications have focused on improving the high-temperature mechanical performance of high-Cr ferritic–martensitic (FM) steels [20,21,22] and advanced oxide-dispersion strengthened alloys [30,31,32,33]. The underlying principle guiding current processing approaches emphasizes increasing radiation resistance by incorporating more defect recombination sites or enhancing defect sink strength [24,25,26,27,28]. Thus, refining lath structures and precipitates through rapid quenching and relatively low-temperature tempering are key methods explored in developing new TMP routes. By investigating these approaches, this study aims to identify an optimized thermal processing method for two 12Cr steels with traditional and nitrogen-enhanced chemistries to significantly boost high-temperature strength, fracture toughness, and overall irradiation resistance. The study utilizes static fracture resistance (J-R) testing and uniaxial tensile testing at selected temperatures to evaluate key TMP conditions and proposes an optimized TMP route for 12Cr steels in nuclear reactor applications.

## 2. Experimental

### 2.1. Compositions and Thermodynamics Guidance

To evaluate the effects of thermomechanical treatments on the microstructural and mechanical characteristics of HT9 steels, two 12Cr–1MoVW alloys were selected and thermomechanically treated in various conditions. Table 1 lists the chemical compositions of these alloys, where it is noted that the HT9 heat-4 is an N-doped version of the standard HT9 steel (Heat-3). The starting HT9 materials of ~10.5 mm thick plates were produced via the traditional TMP for HT9 steel: the HT9 plates with respective compositions were cross-rolled twice at 1000 °C (once perpendicular and once parallel to the original extrusion direction), normalized at 1040 °C for 30 min and air-cooled, and then tempered at 760 °C for 1 h (followed by air-cooling) [34]. As the first step of the TMP schedule, the ~10.5 mm thick initial material plates were normalized at 1100 °C for 30 min, hot-rolled to 2.5–4 mm thick plates, and cut to 40 mm wide, 50 mm long coupons for the final thermal processes described in the following section.

In this research, new thermomechanical processing routes incorporating rapid quenching and controlled tempering steps were selected and applied to two Fe–12Cr alloys to produce ultrafine microstructures with nanoscale laths and finely dispersed carbide particles in high-Cr steels. Various single- and double-tempering conditions were designed to encompass all levels of precipitation, from untempered to fully tempered states. Phase equilibrium diagrams during annealing were generated using thermodynamic calculations to guide the TMP scheduling and understand minor precipitation phases in the 12Cr steel alloys [34,35]. The study employed FactSage 7.0 software for multi-phase thermodynamic calculations, and the results are summarized below.

Figure 1 illustrates the phase equilibrium calculations, focusing on minor phase total contents in two HT9 alloys. Figure 1a depicts the formation of M_23_C_6_, σ phase, Laves phase, and ferrite (BCC) in the low-temperature region <~800 °C. In Figure 1b, the alloy with higher nitrogen content shows a small amount of FCC phase (mostly CrN) persisting in the low-temperature range, diminishing with increasing temperature. Apart from the presence of CrN, overall phase equilibrium behavior between the two HT9 steels remaineds largely similar, with slight differences observed in the intercritical region and high-temperature ferrite region.

Among the minor phases, M_23_C_6_ carbides remained stable up to about 800 °C, rapidly decreasing thereafter and completely dissolving just below 1050 °C [35]. Carbide dissolution may accelerate in less stable microstructures such as those in as-quenched and under-tempered conditions. Temperatures exceeding 800 °C during tempering could lead to over-tempering or microstructure coarsening. Another significant minor phase was the σ-phase, which peaked at 380 °C and disappeared above 480 °C, suggesting that achieving good mechanical properties may be challenging at tempering temperatures well below 500 °C. The predominant intermetallic compounds included Fe_8_Cr_22_ and Fe_8_Cr_4_Mn_18_, with Fe_26_Cr_4_ being another major phase within the sigma solution [35]. Additionally, the Laves phase was mostly present at relatively low temperatures < 600 °C, with its fraction higher at lower temperatures (~1 wt.% at 300 °C) and decreasing to nearly 0% above 600 °C [35]. The Laves phase consisted mainly of compositions such as Fe_2_W, Fe_2_Mo, Cr_2_W, and Cr_2_Mo, with W having the highest weight fraction, followed by Fe, Mo, and Cr among the major alloying elements.

While aiming for an incompletely tempered microstructure to achieve finer microstructures and higher strength, efforts were made to minimize the formation of brittle intermetallic phases such as the σ-phase and Laves phase. As shown in Figure 1, the weight percent of the σ phase ranged from 1.4% to 4.2% below ~480 °C, peaking at ~380 °C. Such significant amounts of the σ phase formed during tempering may contribute to reduced ductility and fracture toughness when tempered at relatively low temperatures < 600 °C. Additional tempering above 500 °C could dissolve the σ phase and transition the matrix into an oversaturated solid solution state, potentially mitigating any significant reduction in ductility or fracture toughness. Other minor phases, such as the Laves phase in both alloys and the FCC (CrN) phase in heat-4, maintained negligible to low fractions (≤1 wt.%) across tempering temperatures. These phases were unlikely to grow into excessively large particles at such low temperatures, thus exerting minimal influence on mechanical properties.

These calculations suggest that tempering should ideally occur within the temperature range of 500–800 °C to minimize the formation of brittle phases, except for very fine carbides or nitrides, while achieving a favorable balance of strength, ductility, and fracture toughness.

### 2.2. Heat Treatment Schedules

Figure 2 illustrates the schematics of various heat treatment routes designed for this research, encompassing different combinations of post-quench treatments. Our goal was to enhance the mechanical performance of thin core components (e.g., approximately ~0.5 mm for fuel cladding and ~3 mm for fuel ducts in sodium-cooled reactors). Thin coupons (2.5–4 mm) were used to achieve rapid cooling rates of 100 °C/s or higher, enabling the production of a nanoscale martensitic lath structure during quenching. In the final stage, efforts were made to minimize the degree of tempering compared to traditional full tempering practices, aiming to refine precipitates [23] within the ultrafine lath structure obtained through rapid quenching. This approach seeks to achieve both higher strength and fracture toughness.

A series of post-rolling thermal processes, including normalization (to dissolve precipitates in the austenite region), rapid quenching in water, and single-step or two-step tempering, are summarized in Table 2. Initially, the coupons measuring 40 mm × 50 mm were hot-rolled at 1100 °C, involving multiple heating–rolling cycles to achieve a thickness reduction of 2.5–4 mm and to eliminate initial microstructural features. These coupons were used in their as-rolled (AR) or as-normalized states for subsequent heat treatments.

The next step in thermomechanical processing (TMP) involved normalizing at 1070 °C for 1 h to create a martensitic structure with very fine laths, followed by rapid quenching in agitated water (WQ). Subsequently, the as-quenched coupons underwent tempering before being machined into specimens for mechanical testing. The tempering treatments included three groups: (i) single-step tempering for 1 h at temperatures ranging from 300 to 750 °C, comparable to typical processes for most ferritic–martensitic steels, including HT9 steels [9,21,22]; (ii) two-step tempering involving lower-temperature (300–600 °C) tempering for 1 h and higher-temperature (650 °C) tempering for 0.25 or 0.5 h; and (iii) two-step tempering with lower-temperature (300–600 °C) tempering for 1 h and higher-temperature (750 °C) tempering for 0.25 or 0.5 h. All tempering steps were followed by air-cooling, aimed at generating fine carbides and other precipitates in the quenched martensite (possibly with a small amount of ferrite) structure.

Following thermomechanical processing, subsize tensile specimens (SS-3) and miniature bend bar fracture specimens were machined from the treated coupons. As shown in Table 2, these specimens were tested in the as-rolled (AR) condition, as-water quenched (WQ) condition, six single-tempered conditions, or ten double-tempered conditions. For each TMP condition, both uniaxial tensile testing and static fracture testing were conducted at temperatures of 25, 200, 300, 400, 500, and/or 600 °C.

### 2.3. Mechanical Testing

To obtain mechanical performance data for comparing HT9 steels treated under various conditions, we conducted uniaxial tensile testing and fracture resistance (J-R) testing for each alloy and processing condition at the same test temperatures. Note that only one test was performed per alloy, processing condition, and test temperature, so statistical analysis for each data point was not possible. Although we could not conduct multiple tests under each condition due to the extensive number of tests required, we chose to test each material at various temperatures to identify trends in temperature dependence and detect potential outliers.

Uniaxial tensile tests were conducted using SS-3 specimens from two HT9 steels subjected to various heat treatments, including rapid water quenching and one-step or two-step tempering (see Table 2). The SS-3 subsize tensile specimens featured a flat gauge section measuring 7.62 mm in length, 1.52 mm in width, and 0.76 mm in thickness, with a total length of 25.4 mm and a head width of 5 mm. Tensile tests were carried out at temperatures of 25, 200, 300, 400, 500, and 600 °C using a screw-driven testing machine equipped with a high-temperature furnace, at a displacement rate of 0.5 mm/min, corresponding to a nominal strain rate of 0.0011/s. In all tensile tests, the loading direction was aligned with the hot-rolling direction (i.e., L-direction). The ASTM standard procedure for tensile testing of metallic materials [36] was followed to determine engineering strength and ductility parameters, including yield strength (YS), ultimate tensile strength (UTS), uniform elongation (UE), and total elongation (TE), based on raw tensile load-displacement data and specimen dimensions.

Static fracture resistance (J-R) testing was performed using miniature bend bar specimens in an electromagnetically driven testing machine equipped with an environmentally controlled high-temperature furnace using argon gas [18]. The fracture specimens, measuring 14 mm in length, 4 mm in width, and 3 mm in thickness, were single-edge bend bars (SEBs) with a 1.5 mm deep notch and a 0.45 mm (15% of thickness) deep side groove on each side, oriented in the L-T direction (loading along the rolling direction, crack propagation perpendicular to it). Pre-cracking involved cyclic loading at 300 ± 200 N at 10 Hz until the machined notch extended by 0.3–0.7 mm. After confirming the desired crack length extension, the load amplitude was reduced by 50%, and an additional 10,000 cycles were applied to sharpen the crack tip. The nominal crack length-to-specimen width ratio (a/W) was approximately 0.5 before J-R testing.

Fracture resistance testing (J-R) was conducted quasi-statically at a crosshead speed of 0.3 mm/min, maintaining temperature control within ±3 °C. For each TMP condition, J-R testing was performed at temperatures of 25, 200, 300, 400, 500, and 600 °C, either in argon gas or air, using a displacement-controlled three-point bending setup following ASTM Standard E1820-09 [37]. In the three-point bending fracture testing, the loading direction was aligned with the crack propagation direction (i.e., the T-direction in hot rolling). As a result, the principal tensile stress at the crack tip was oriented along the hot-rolling direction (i.e., the L-direction), which matched the loading direction in tensile testing. The construction of the J-R curves utilized the curve normalization method [18,37], where monotonic load–displacement curves were recorded without elastic loading–unloading cycles. Each test concluded when the load measurement reached a maximum and then decreased to about 55% of the maximum load unless a catastrophic failure occurred earlier. Heat-tinting was applied post-test during slow cooling to mark the final crack length before complete specimen separation. Initial and final crack lengths were measured from optical photographs and used to calculate the J-R data.

In the data analysis procedure to construct the J-Resistance curve (J-integral versus Δa curve), crack lengths during crack growth were determined between the measured initial and final crack lengths using the curve normalization method, as described in ASTM Standard E1820 (section A15). While the detailed calculation procedure was simplified in this study by eliminating the external clip-on gage typically used to measure precise crack mouth opening displacement [18,37], two experimental datasets were essential for constructing each J-R curve: the load-displacement curve and the initial and final crack lengths. Interim fracture toughness (J_Q_) values were derived at the intersection of the J-R curve and the 0.2 mm offset lines of the blunting line. These J_Q_ values were subsequently converted into stress intensity factor (K_JQ_) values. It is noted that the fracture toughness data (K_JQ_) presented in this study are interim values due to limitations imposed by specimen size and thickness, which prevent full compliance with all requirements for validated fracture toughness (K_IC_) [33].

## 3. Results

### 3.1. Stress-Strain Behavior and Strength of HT9 Steels in Various TMP Conditions

Two sets of engineering stress-strain curves are shown in Figure 3a,b for the lowest and highest test temperatures (RT and 600 °C), respectively. These curves were chosen to illustrate the stress-strain behaviors under key TMP conditions. The water-quenched (WQ) condition represents the extremely hardened state of the HT9 steels, while WQ-750 °C serves as the reference (fully tempered) condition. Additionally, WQ-650 °C and WQ-600 °C–650 °C may correspond to optimized conditions, as will be discussed in the following sections. Other TMP conditions tested in this research are expected to display properties within the ranges indicated by these selected cases.

Since more detailed mechanical behaviors with data on various properties will be discussed in the following sections, this summary focuses only on the overall deformation behaviors as depicted by the stress-strain curves. First, both HT9 steels in the water-quenched (WQ) condition exhibited very high flow stresses, exceeding 1 GPa at room temperature, but with reduced ductility. Specifically, HT9 steel heat-3 (represented by the 3WQ curve) showed nearly embrittled behavior at room temperature, failing either before or very close to reaching the ultimate tensile strength or the onset of necking, as seen in Figure 3a. Second, HT9 steel heat-4 demonstrated relatively lower flow stress in both the as-quenched and reference (fully tempered) conditions, while the other two conditions showed higher flow stress compared to heat-3. This trend is consistent with the stress-strain curves obtained from testing at 600 °C. Third, regardless of the test temperature, the widths of the stress-strain curves (indicating total ductility) were similar for samples tempered at either 650 °C or 750 °C. This suggests that the tempering conditions have a comparable impact on ductility at these temperatures. Finally, Figure 3b shows that both HT9 heats initiate plastic instability (necking) very early, within a few percent of strain, but continue to deform significantly and exhibit considerable necking ductility. As will be discussed in detail later with the uniform and total elongation data, this high-temperature behavior—characterized by prompt plastic instability and large necking ductility—appears to be a notable feature of HT9 steels.

The strength data for HT9 steels after various TMP routes are compiled in Figure 4 and Figure 5, showing YS and UTS versus test temperature plots. As is typical for phase-transformable ferritic steels, a wide range of strengths was observed, depending largely on the thermal processing route, particularly the degree of tempering. Among room-temperature data, the highest YS values were found in the as-rolled (AR) condition, approximately 1320 MPa and 1470 MPa for HT9 heat-3 and heat-4, respectively. It is noteworthy that both alloys, either before tempering or after a single tempering below 600 °C, exhibited yield stresses above 1 GPa. Additionally, certain low-temperature tempering treatments resulted in YS values higher than those of the as-quenched materials.

Two-step tempering treatments at 650 °C/0.5 h and 750 °C/0.5 h resulted in a narrow range of both YS and UTS values for the respective alloys. These strength data closely matched those after single tempering at 650 °C and 750 °C for 1 h, indicating that the final tempering temperature is the primary factor controlling the strength of HT9 steels, provided the tempering duration is sufficient for near equilibrium (i.e., 0.5 h to 1 h). The significant effect of tempering temperature on strength suggests that traditional tempering around 750 °C is adequate for inducing the desired tempering effects due to high internal residual stress in the rapidly quenched microstructure, facilitating precipitation and stress relaxation. Temperatures lower than 750 °C, such as 650 °C for one hour, also proved effective in achieving the desired tempering effects.

Furthermore, it was observed that the tempering effect at a high temperature (750 °C) was more pronounced in HT9 heat-3 compared to heat-4, reflected in different YS ranges (~600 MPa for heat-3 versus 440–520 MPa for heat-4 at room temperature). Overall, HT9 heat-4 demonstrated a broader range of strength for the same variety in TMP routes.

The UTS data of HT9 steels, shown in Figure 5, exhibited behavior similar to the YS data but generally about 20% higher over the temperature range up to 500 °C. Both TS and UTS monotonically decreased with test temperature at similar rates, regardless of alloy composition and TMP route. Notably, the decrease in UTS between 500 °C and 600 °C was more pronounced than that of YS, leading to narrower ranges for both at 600 °C, with UTS values only about 5% higher than YS values on average. This indicates a shift from uniform deformation to more significant necking deformation, which is less sensitive to detailed changes in mechanical properties.

Regarding nitrogen doping, it appears to lower the strength of HT9 steel under certain TMP conditions. Specifically, the HT9 heat-4 with higher nitrogen content (0.044 wt.%) exhibited significantly lower strength compared to the low-nitrogen HT9 heat-3 across the entire test temperature range, except in specific processing routes where heat-4 showed a wider range and sometimes higher strength levels (e.g., as-rolled steel) compared to heat-3. In conclusion, while nitrogen addition affects strength non-uniformly across different TMP conditions, it may also increase ductility. However, the precise mechanisms underlying these strength changes due to nitrogen are not fully understood, warranting further investigation into its role in austenite formation below the calculated A3 temperature and in the total amount of precipitates formed and dissolved during heat treatment.

### 3.2. Ductility of HT9 Steels in Various TMP Conditions

The temperature dependence of ductility for the two HT9 steels is summarized in Figure 6 and Figure 7, for UE and TE, respectively. Unlike the monotonic temperature dependence of YS and UTS, ductility parameters exhibit complex dependencies that vary based on processing route and material strength. Generally, the rankings of room-temperature (RT) strength parameters are inversely reflected in ductility parameters, with temperature dependence curves showing either a monotonically decreasing line, a curve with a maximum, or a curve with a minimum across the test temperature range.

Figure 6a,b illustrates that UE data range widely from 1% to 9% at RT. Except for conditions before or after single tempering at 300 °C and 400 °C (AR, WQ, WQ-300 °C, and WQ-400 °C), UE decreased nearly linearly with test temperature, dropping below 2% at 600 °C. This decline may be attributed to the reduced strain-hardening capability of highly tempered HT9 steels at higher temperatures, where carbides and boundaries play less effective strengthening roles.

Conversely, materials with minimal tempering exhibit a rapid increase followed by a gradual decrease in UE above RT, with peak values occurring between 300 °C and 500 °C. Such low UE values at lower test temperatures (RT and 200 °C), despite their high strengths, can lead to early plastic instability and low uniform ductility. This is due to high initial flow stress, which leads the material to meet the instability criterion prematurely. As the temperature surpasses the point of maximum UE, ductility rapidly diminishes, and this is particularly evident at 600 °C. Consequently, the lowest UE within each TMP condition typically occurs at the highest test temperature. The primary strengthening features in these lightly tempered microstructures are likely ultrafine martensitic or ferritic laths, possibly with fine carbide or sigma phase particles [35]. These features, lacking the pinning effect of precipitates but with high residual stresses, may hinder dislocation glide above 400 or 500 °C, resulting in low UEs (≤~1%).

Figure 7a,b summarizes the total elongation (TE) data for the two HT9 heats, respectively. TE represents an alloy’s ability to resist failure and can differ significantly from uniform ductility. Ferritic–martensitic steels typically absorb more plasticity in necking deformation than in uniform deformation, and the TE data encompass both necking and final fracture information. Figure 7 confirms that, aside from the near-embrittled case (3WQ at RT), all processing routes achieved at least 5% TE, with the majority exceeding 10% TE. Notably, TE values at the highest test temperature of 600 °C rivaled those at RT, predominantly falling within 16–18% for heat-3 alloys and 10–20% for heat-4 alloys.

Meanwhile, the temperature dependencies of less-tempered materials were non-monotonic and differed from those of more-tempered materials. For instance, the TE of heat-3 suggested near embrittlement at RT and peaked at 300 °C, whereas heat-4 recorded a minimum at 200 °C and increased with test temperature up to the highest point. Compared to UE behavior, the TE data exhibit narrower bands, with most TEs significantly higher than UEs, affirming that tensile deformation in HT9 steels spends more than 10% elongation in necking mode. Importantly, the steep decline in high-temperature regions observed in UE versus test temperature curves was absent in TE temperature dependence, indicating that no embrittlement mechanism operated at elevated temperatures across these alloys, despite various TMPs.

Another distinct aspect of TE, differing from UE, is the clear formation of a ductility minimum in its temperature dependence curve. Except for less-tempered conditions (e.g., AR, WQ, WQ-300 °C, or WQ-400 °C), resulting in very low TE at temperatures < 300 °C, both HT9 steels exhibited high elongation at RT (12–20%), declining to a minimum around 400 ± 100 °C, then rising again in a U-shaped curve with increasing temperature. This ductility pattern, showing a minimum point, is attributed to the dynamic strain aging (DSA) phenomenon [38,39,40,41]. DSA reduces dislocation mobility due to repeated pinning–unpinning motions by interstitial atoms, primarily carbon, typically leading to decreased ductility and toughness in metallic materials. Notably, the minimum fracture toughness of HT9 steel (INL) aligns with this temperature at 400 °C [14]. The consistent occurrence of DSA effects around ~400 °C in multiple HT9 alloys suggests a fundamental phenomenon influenced by light elements such as carbon and nitrogen [19].

### 3.3. Effect of TMP Route on the Static Fracture Toughness of HT9 Steels

Figure 8 and Figure 9 present fracture toughness (K_JQ_) versus test temperature data ranging from 25 °C to 600 °C for the 12Cr steel heats 3 and 4, respectively. The objective of this testing and evaluation campaign was to identify a thermal–mechanical processing (TMP) route capable of enhancing both the strength and fracture toughness of HT9 steels, or at least one of these properties without compromising the other. This improvement is particularly crucial for the high-temperature range exceeding 550 °C, which aligns with the anticipated thermal capabilities of future fast reactors. Existing fracture toughness data for FFTF and INL HT9 steels are also included in Figure 8 and Figure 9 for comparative purposes. A reasonable criterion for selecting a new TMP should ensure that neither fracture toughness nor strength decreases compared to these reference materials when implementing the new schedules.

The traditionally tempered Fe–12Cr ferritic–martensitic steels, including FFTF and INL HT9, typically exhibit a monotonic decrease in fracture toughness with increasing test temperature above room temperature or the upper shelf region [14,19]. This decline is primarily due to the steep reduction in strength at higher temperatures. However, within the elevated temperature range of 200 °C to 400 °C, there is often an additional reduction or local minimum in fracture toughness. This phenomenon, i.e., the effect of DSA also observed in the ductility data, occurs due to the hindrance of dislocation glide by interstitial elements such as carbon and nitrogen, which follow the stress fields of moving dislocations [38,39,40,41]. Although the impact of DSA on reducing toughness is generally modest in HT9 steels [14], the fracture toughness versus temperature curve still exhibits a minimum or low-value region before rising at higher temperatures. This subsequent increase in fracture toughness at temperatures typically above 400 °C is primarily attributed to increased ductility beyond the DSA-affected region. Therefore, the temperature dependence of fracture toughness in fully tempered HT9 steels above room temperature typically shows either a monotonic decrease or a decrease followed by an increase with test temperature.

Figure 8a illustrates that room temperature fracture toughness is generally high (>~200 MPa√m), except for alloys before tempering or after low-temperature tempering (e.g., AR, WQ-300 °C, and WQ-400 °C conditions). Above room temperature, the fracture toughness dependence on temperature was influenced by the tempering degree. K_JQ_ values for HT9 steel heat-3 after WQ-400 °C, WQ-300 °C–650 °C, WQ-400 °C–650 °C, and WQ-600 °C–650 °C processes showed a monotonic decrease with test temperature, whereas those after the TMT routes of WQ-400 °C, WQ-500 °C, WQ-600 °C, and WQ-500 °C–650 °C initially decreased and then increased with test temperature. The AR condition resulted in low room-temperature fracture toughness, but this increased monotonically with temperature thereafter, suggesting that the toughness of this non-tempered material was highly dependent on its total elongation (ductility). Notably, cases like 3WQ-500 °C and 3WQ-600 °C demonstrated significantly improved K_JQ_ values in the range of 250–310 MPa√m at 600 °C, which compares favorably to the typical K_JQ_ range of 150–200 MPa√m for fully tempered HT9 steels (e.g., FFTF and INL alloys). Given that reference materials commonly show a monotonic decrease in fracture toughness with test temperature, the observed decrease–increase cycle in fracture toughness under some under-tempered conditions provides promising evidence of improved mechanical performance.

Figure 8b compares fracture toughness data for HT9 heat-3 after two-step tempering, with the second tempering performed at either 650 °C or 750 °C for 0.5 h or 0.25 h (denoted as ‘S’). Regardless of the detailed tempering conditions, all combinations of tempering steps yielded comparable or higher fracture toughness over the test temperature range compared to reference HT9 steels. Notably, the materials subjected to two tempering treatments, such as 500 °C/1 h–650 °C/0.25 h and 600 °C/1 h–750 °C/0.25 h, maintained fracture toughness well above 200 MPa√m in the high-temperature range (e.g., 278 and 312 MPa√m at 600 °C). Particularly desirable was the observation that the fracture toughness of these materials could increase with test temperature in the 500–600 °C range, while materials subjected to other tempering treatments, including reference cases, experienced decreasing fracture toughness.

In Figure 9, the fracture toughness data of HT9 heat-4 after various thermomechanical processing (TMP) treatments are compared with those of reference HT9 steels. Figure 9a clearly shows that the overall K_JQ_ range of HT9 heat-4 after hot rolling alone or any single-step tempering was slightly lower than the reference data. Additionally, it was observed that the fracture toughness of HT9 heat-4 was generally lower than that of HT9 heat-3. Samples tempered at or above 600 °C exhibited either a monotonic decrease in fracture toughness with test temperature from highest values at room temperature or a decrease in the low-temperature region followed by a slight increase above 300 °C. Severely under-tempered cases showed a temperature-transition behavior (a steep increase with temperature) at ≤300 °C. K_JQ_ values without tempering and after relatively low degrees of tempering (AR, WA-300 °C, WQ-400 °C, and WQ-500 °C) increased with test temperature from a low toughness range of 60–80 MPa√m at room temperature. It is evident that the new TMPs with single tempering applied to heat-4 resulted in little improvement in fracture toughness.

Figure 9b presents the fracture toughness data of heat-4 after two-step tempering. The two two-step tempering routes, WQ-500 °C–650 °C and WQ-600 °C–750 °C, yielded fracture toughness similar to that of the reference materials. No other tempering route noticeably improved fracture toughness with the HT9 alloy heat-4 composition. However, it should be noted that the strength of heat-4 with either of the two-step tempering treatments was significantly higher than that of the fully tempered HT9 steels. At least, the two thermal treatments, WQ-500 °C–650 °C and WQ-600 °C–750 °C, achieved significant gains in strength without sacrificing fracture toughness, indicating an improvement in mechanical performance.

Furthermore, the fracture toughness data described in Figure 9 suggest that no clear improvement in fracture resistance can be achieved with HT9 steel with nitrogen addition. In fact, the discussion concludes that the addition of nitrogen (~0.044%) to the original HT9 composition is mostly detrimental to the alloy’s fracture resistance. However, it is noted that the effect of nitrogen addition on fracture resistance may differ after irradiation, as a controlled amount of nitrogen atoms can reduce the accumulation of radiation damage [42,43]. The cause of this effect requires further investigation, but the present K_JQ_ dataset provides evidence that nitrogen addition can form a nitrogen-stabilized second phase, which might not contribute to ductility or enhance fracture resistance.

## 4. Discussion on Microstructure and Mechanical Properties

To understand the general microstructural evolution of the two HT9 steels during processing, martensite lath structure and carbide precipitation were examined using transmission electron microscopy (TEM) in two extreme conditions: the as-quenched state and after quenching followed by tempering at 750 °C. These conditions represent the extremes of the steels, maximizing strength and ductility, respectively, and denote the least and most relaxed states in the current processing development scope. An optimal thermomechanical treatment (TMT) condition for achieving the best combination of mechanical properties was expected to lie between these extremes, necessitating further microscopy analysis for detailed comparison.

The TEM images in Figure 10 confirm that finer and generally more planar laths were formed with the traditional composition (heat-3). The as-quenched microstructure consisted predominantly of lath martensite or subgrain structures with high-density dislocations. After tempering at 750 °C for 1 h, carbides and distinct lath and subgrain boundaries became evident. The lath thicknesses, measured as the intercept lengths, were approximately 280 nm and 470 nm for heat-3 and heat-4, respectively, in the as-quenched condition, as shown in Figure 10a,b. After tempering, these measurements decreased slightly to about 250 nm and 460 nm, respectively, because the tempering treatment made the lath boundaries more distinct, as illustrated in Figure 10c,d. The morphology and size of the laths (or ferrite subgrains) suggest that the hardenability required to achieve a fine quenched structure is lower in heat-4 with nitrogen addition. As a result, the heat-4 alloy exhibited relatively lower strength under many conditions and showed only limited improvement in fracture toughness. This outcome likely represents a primary adverse effect of nitrogen addition, which contradicts the original intent of the alloy development, aimed at increasing precipitates and thereby enhancing its properties. Future studies should focus on designing improved TMT routes that promote the precipitation of nanoparticles, including nitrides.

Changes in the distribution of chemical elements during thermomechanical processing were investigated using a scanning transmission electron microscope (STEM) and are displayed in Figure 11 and Figure 12 for the original composition and nitrogen-doped HT9 steels, respectively. STEM-EDS analysis was performed for both the as-quenched condition and the fully tempered condition to compare elemental distribution before and after carbide precipitation through tempering. Figure 11a and Figure 12a display the elemental maps of the as-quenched microstructures, which demonstrate that rapid quenching (thin coupons cooled in water) created a pristine and metastable state in both HT9 alloys, showing neither precipitation nor segregation before tempering. In these as-quenched steels, consisting of martensite or ferrite laths, minimal boundary segregation of alloy elements was observed.

In the fully-tempered HT9 steel heat-3, as shown in Figure 11b,c, carbide particles with enriched or depleted elements were clearly visible. Elements such as Cr, Mo, and V were enriched in the carbide particles formed during tempering at 750 °C, while Fe was depleted within these particles. Nickel showed no visible partitioning after tempering, confirming the results of thermodynamic calculations that M_23_C_6_-type precipitation (e.g., Cr_20_Mo_3_C_6_) is the most common carbide in tempered HT9 steels. It was also observed that W and S had some affinity with C and Cr or limited enrichment in or around carbide particles.

In addition, Figure 12b,c indicates that the addition of a small amount of nitrogen (~0.044%) to the HT9 alloy did not substantially alter precipitation behavior during heat treatment steps, including normalization, quenching, and tempering. Similar features of carbide particles with enriched or depleted elements were observed, where carbide-forming elements such as Cr, Mo, and V were enriched in the carbide particles. The EDS maps of heat-3 confirm Fe depletion and no Ni enrichment in the carbide particles during the 750 °C tempering. While overall precipitation behavior was similar for the two HT9 steels, one notable effect observed due to nitrogen addition was the presence of some large carbide particles, approximately 1 μm in length, in HT9 heat-4. These large particles were clearly visible in the EDS maps, showing Cr, Mo, W, C, and S elements, as depicted in Figure 12c. As illustrated in Figure 1, the presence of other phases, such as Laves phases (Fe_2_W, Fe_2_Mo, Cr_2_W, and Cr_2_Mo), intermetallic Fe–Cr or Fe–Mn–Cr compounds, or the Fe–Cr sigma phase [35], was considered negligible in these microstructures formed at 750 °C.

As discussed in earlier sections, although nitrogen (0.044 wt.%) was added to the HT9 alloy with the intention of improving mechanical properties, no improvement in the measured properties was achieved. Instead, our discussion in Section 3.3 concluded that the addition of nitrogen to the original HT9 composition was actually detrimental to the alloy’s fracture resistance. The cause of this result is not well understood, and requires further study. One conjecture for this outcome is that the formation of a small amount of CrN phase in HT9 heat-4 during the 750 °C tempering might be responsible for the decreased fracture toughness. As discussed in Section 2.1 and 3.3, overall phase equilibrium behavior between the two HT9 steels remains largely similar, except for the N-enriched phase prosily formed in heat-4. Finally, it is noted that nitrogen detection was not correctly performed in the STEM analysis, and therefore, the presence of the N-rich phase was not confirmed.

## 5. Summary and Conclusions

This study aimed to identify a new processing route for HT9 steels to enhance their mechanical performance in high-temperature reactor conditions, focusing on improving both high-temperature strength and fracture toughness. Our approach was based on the well-established observation that finer microstructures generally exhibit higher resistance to radiation damage and better overall mechanical properties. Only practical processing routes, involving rapid quenching and single or double tempering, were explored to effectively refine the quenched lath structure and precipitate dispersion. This was aimed at enhancing the in-reactor performance of critical components such as fuel cladding and fuel ducts. Eighteen thermomechanical processes (TMPs) were developed and applied to two Fe–12Cr (HT9) alloys, with mechanical properties evaluated through uniaxial tensile testing and static fracture resistance (J-R) testing across a wide temperature range of 25–600 °C. Below are the key observations and conclusions drawn from this comprehensive testing and evaluation campaign.

[1]The Fe–12Cr alloys exhibited a broad range of strength depending on the thermomechanical processing route, particularly influenced by the tempering conditions. Notably, both HT9 alloys achieved ultrahigh yield stresses above 1 GPa either before tempering or after single tempering below 600 °C. Yield strength (YS) and ultimate tensile strength (UTS) of HT9 steels exhibited a consistent decrease with increasing test temperature, irrespective of alloy composition or TMP routes. The final tempering temperature emerged as the predominant factor influencing the strength of the HT9 steels.[2]The relative strengths between the two HT9 alloys varied significantly across different thermomechanical processing routes, with nitrogen addition generally reducing the strength of HT9 steel under most TMT conditions. The underlying reasons for this strength reduction with nitrogen doping are currently not well understood, necessitating further comprehensive investigation.[3]In contrast to the straightforward temperature dependence observed in YS and UTS, ductility parameters, such as uniform elongation (UE) and total elongation (TE), displayed more complex behavior influenced by the specific processing routes and resulting material strengths. Generally, the rankings of room-temperature strength parameters were approximately reversed in terms of ductility parameters.[4]The room-temperature fracture toughness of HT9 steels was generally high (i.e., >~200 MPa√m), except in cases with no or low-temperature tempering (e.g., AR, WQ-300 °C, WQ-400 °C). The temperature dependence of fracture toughness above room temperature was strongly correlated with the degree of tempering. Overall, the K_JQ_ range of HT9 heat-4 tended to be slightly lower across most TMPs compared to HT9 heat-3.[5]Certain limited tempering processes on HT9 heat-3, such as WQ-500 °C, WQ-600 °C, WQ-500 °C–650 °C-S, and WQ-600 °C–750 °C-S, significantly enhanced fracture toughness (K_JQ_) at 600 °C, achieving values of 250–320 MPa√m. This improvement represents a substantial enhancement compared to the typical K_JQ_ range of 150–200 MPa√m observed in fully tempered HT9 steels. However, the newly tested TMPs on heat-4 showed minimal improvement in fracture toughness, although WQ-500 °C–650 °C and WQ-600 °C–650 °C did achieve considerable additional strengthening without compromising fracture toughness.

## Figures and Tables

**Figure 1 materials-17-03803-f001:**
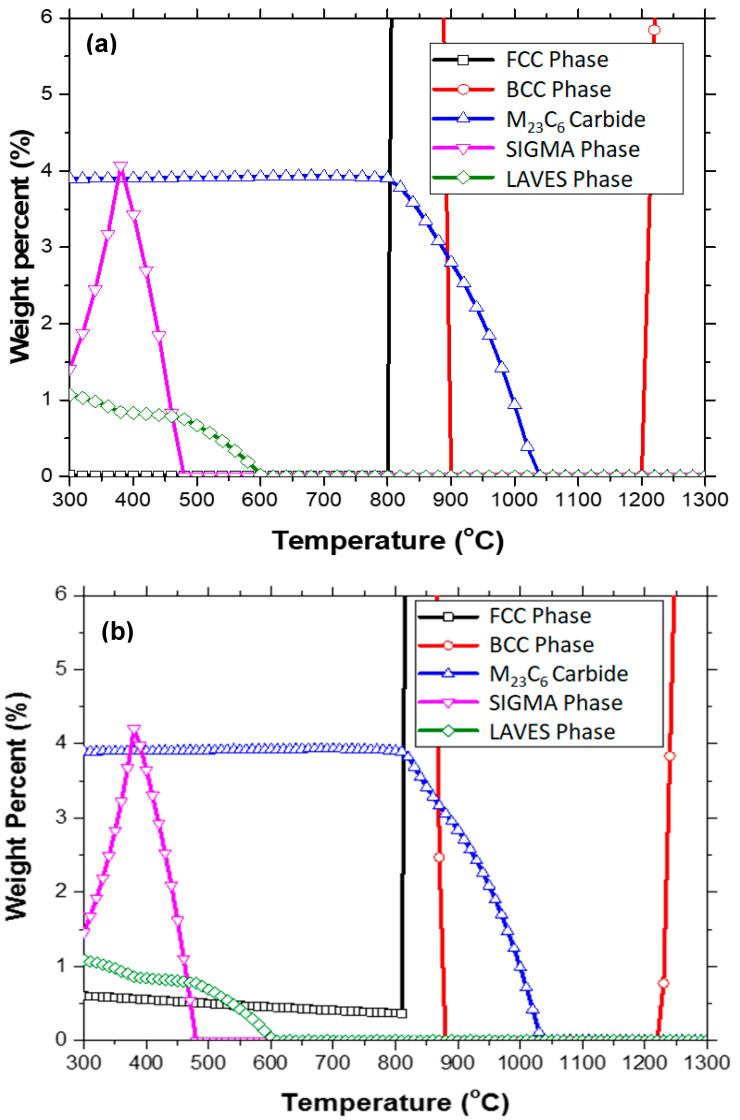
Phase diagram enlarged for minor phases formed in the HT9 alloys: (**a**) heat-3 and (**b**) heat-4 (obtained via thermodynamics simulation).

**Figure 2 materials-17-03803-f002:**
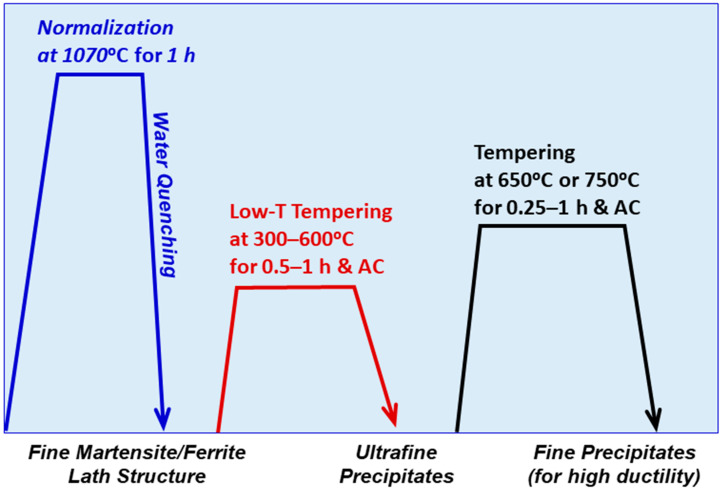
Schematic of possible thermal processing routes for ferritic–martensitic steels: a variety of thermomechanical processing routes can be designed by combining the normalization–quenching step with different single-step and two-step tempering processes.

**Figure 3 materials-17-03803-f003:**
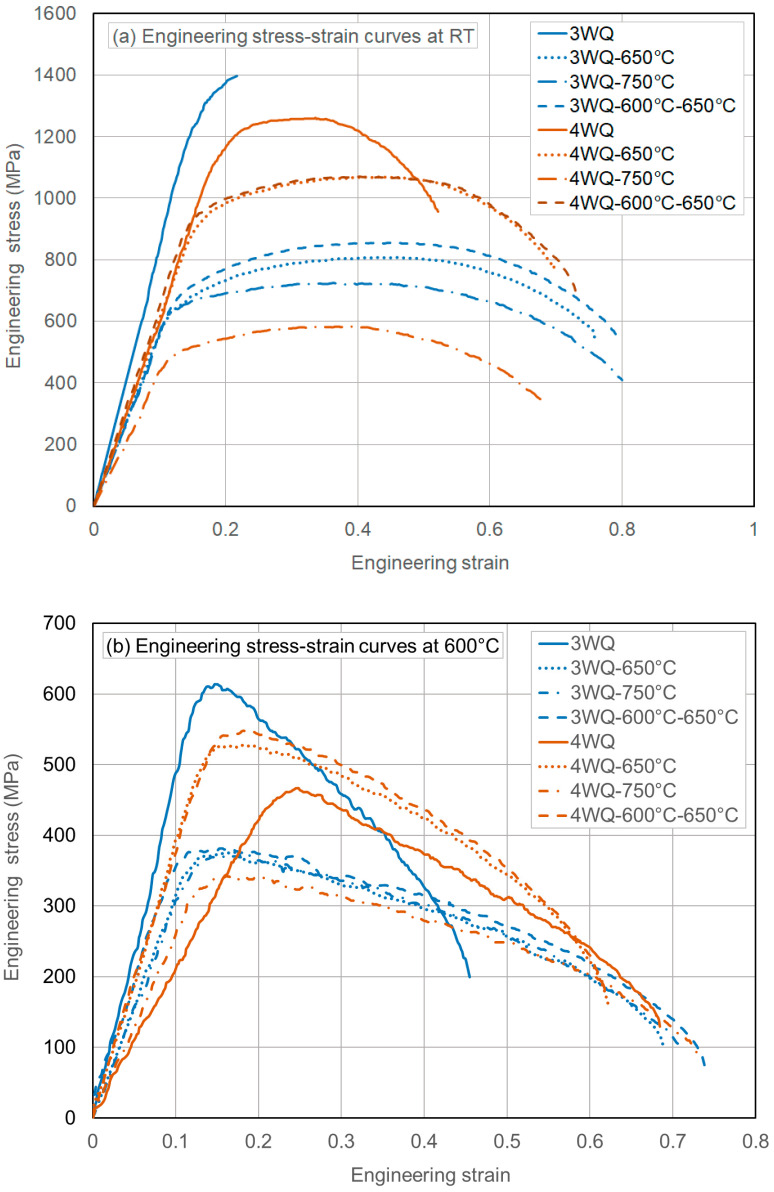
Engineering stress-strain curves of (**a**) HT9 steel heat-3 (low N) and (**b**) HT9 steel heat-4 (high N) in selected thermal treatment conditions.

**Figure 4 materials-17-03803-f004:**
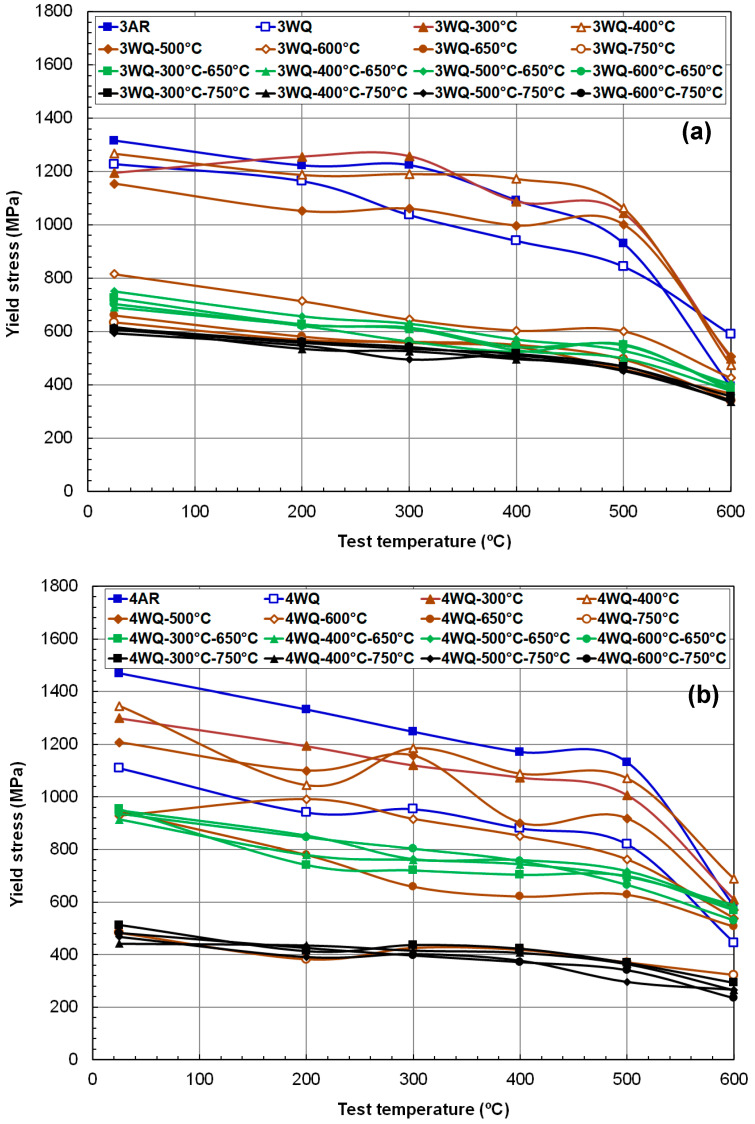
Temperature dependence of yield stress (**a**) in HT9 steel heat-3 (low N) and (**b**) in HT9 steel heat-4 (high N) after various thermal treatment routes.

**Figure 5 materials-17-03803-f005:**
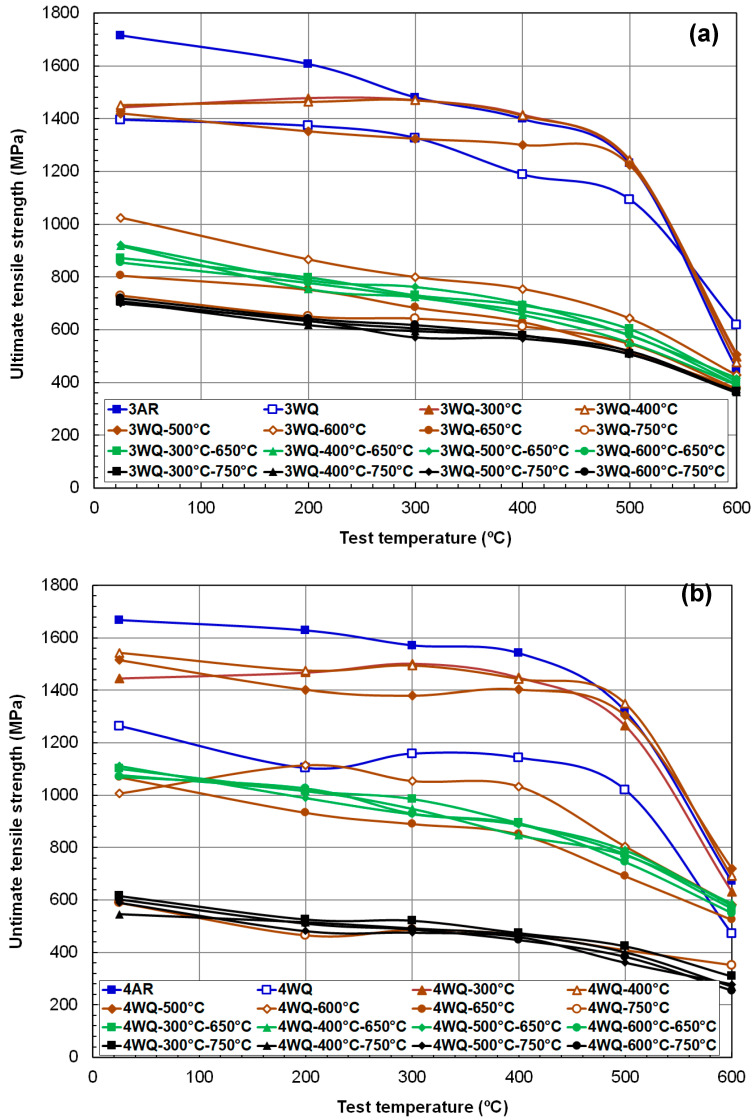
Temperature dependence of ultimate tensile strength (**a**) in HT9 steel heat-3 (low N) and (**b**) in HT9 steel heat-4 (high N) various thermal treatment routes.

**Figure 6 materials-17-03803-f006:**
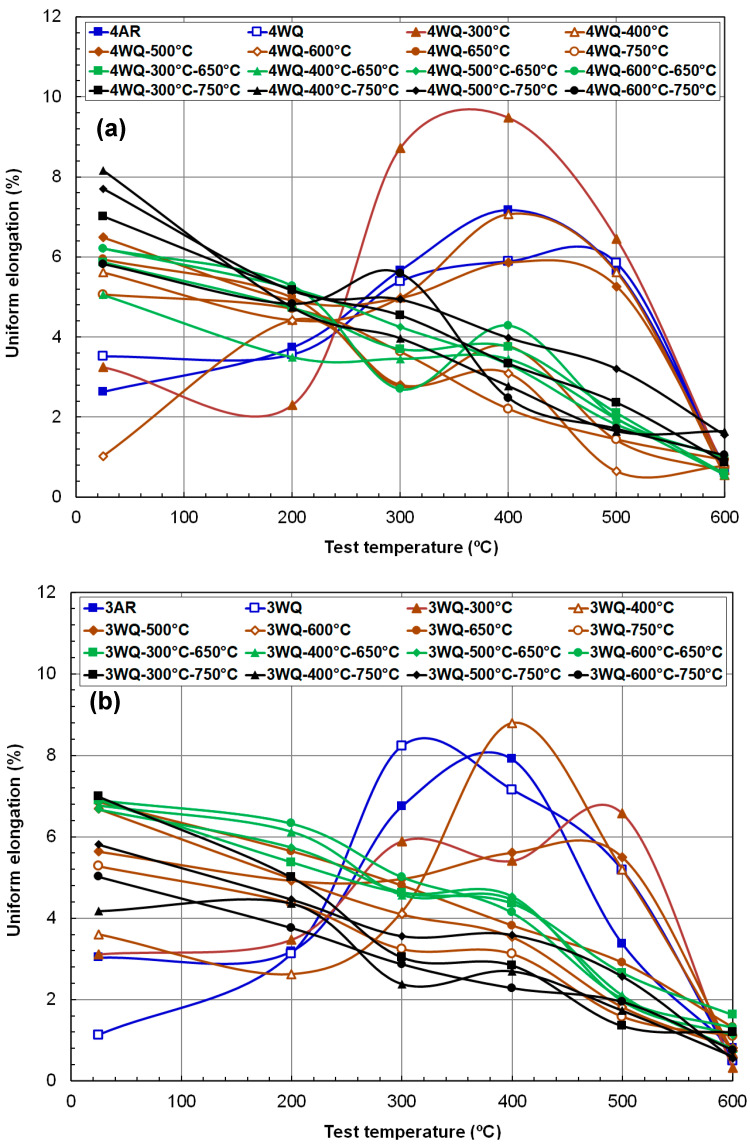
Temperature dependence of uniform elongation (**a**) in HT9 steel heat-3 and (**b**) in HT9 steel heat-4 after various thermal treatment routes.

**Figure 7 materials-17-03803-f007:**
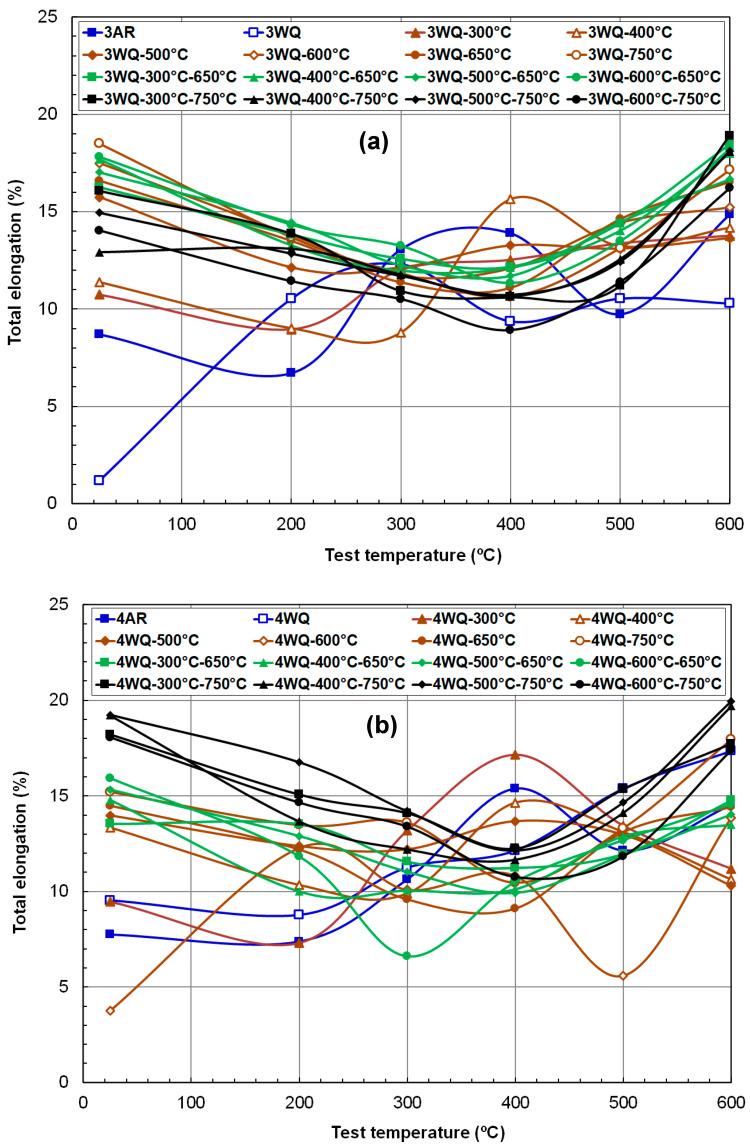
Temperature dependence of total elongation (**a**) in HT9 steel heat-3 and (**b**) in HT9 steel heat-4 after various thermal treatment routes.

**Figure 8 materials-17-03803-f008:**
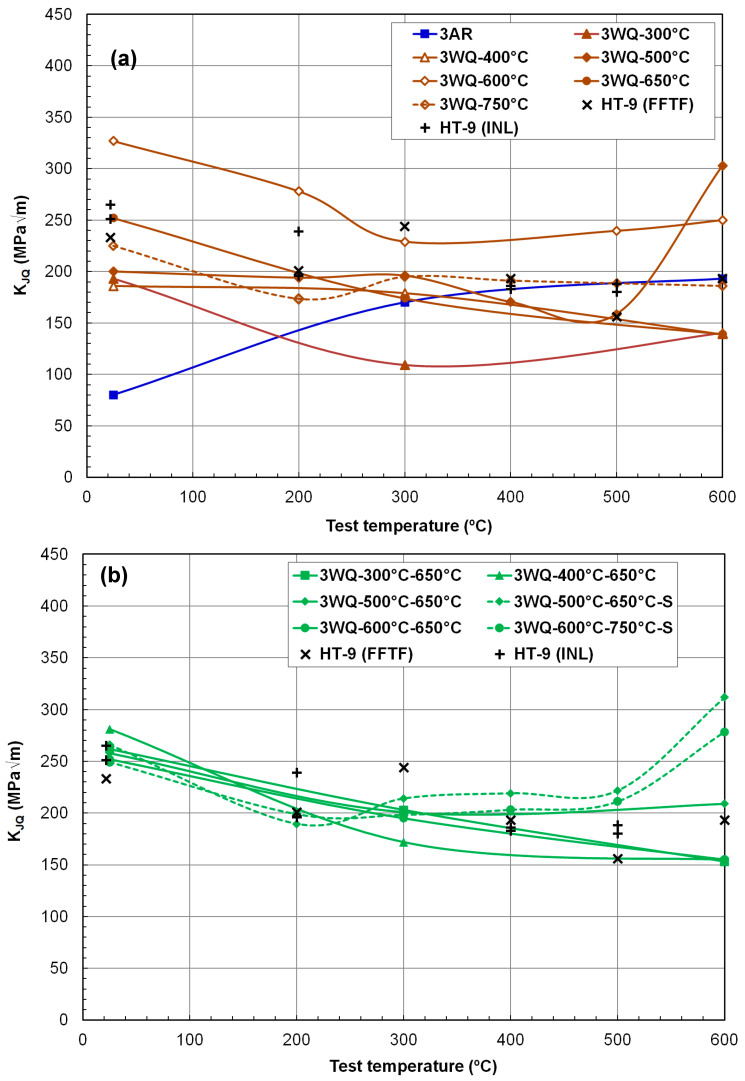
Fracture toughness of HT9 steel heat-3 in various thermomechanical processing conditions including (**a**) as-rolled (AR), water-quenched (WQ), single-tempered (1 h), and (**b**) double-tempered conditions.

**Figure 9 materials-17-03803-f009:**
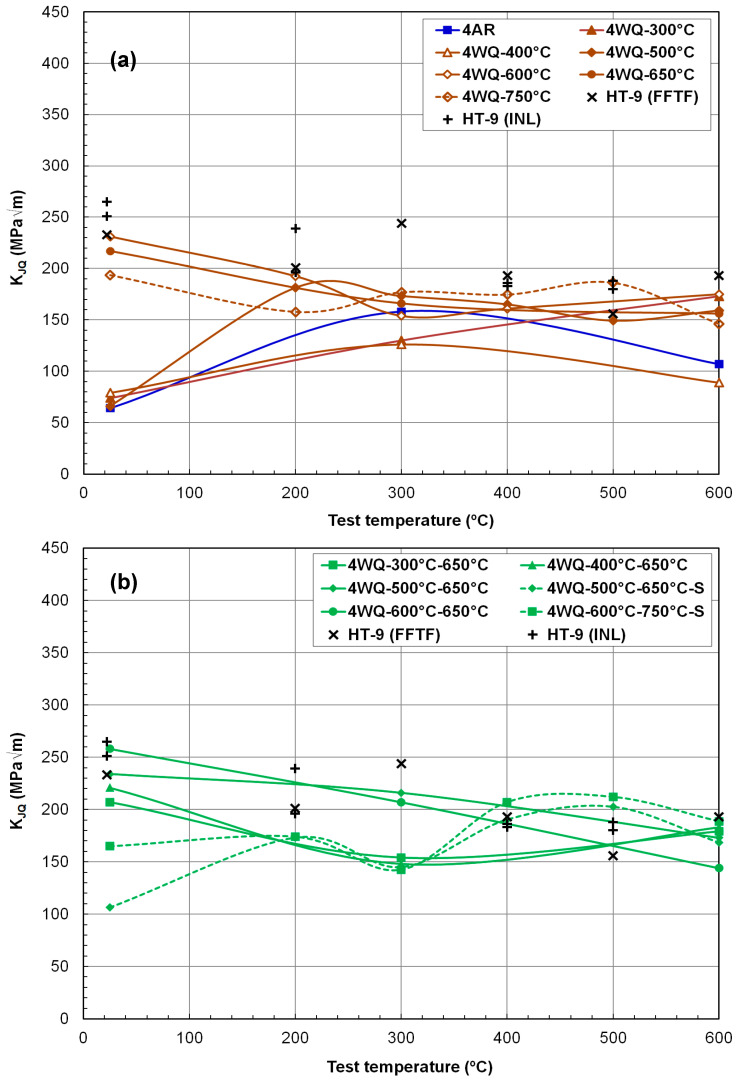
Fracture toughness of HT9 steel heat-4 in various thermomechanical processing conditions including (**a**) as-rolled (AR), water-quenched (WQ), single-tempered (1 h), and (**b**) double-tempered conditions.

**Figure 10 materials-17-03803-f010:**
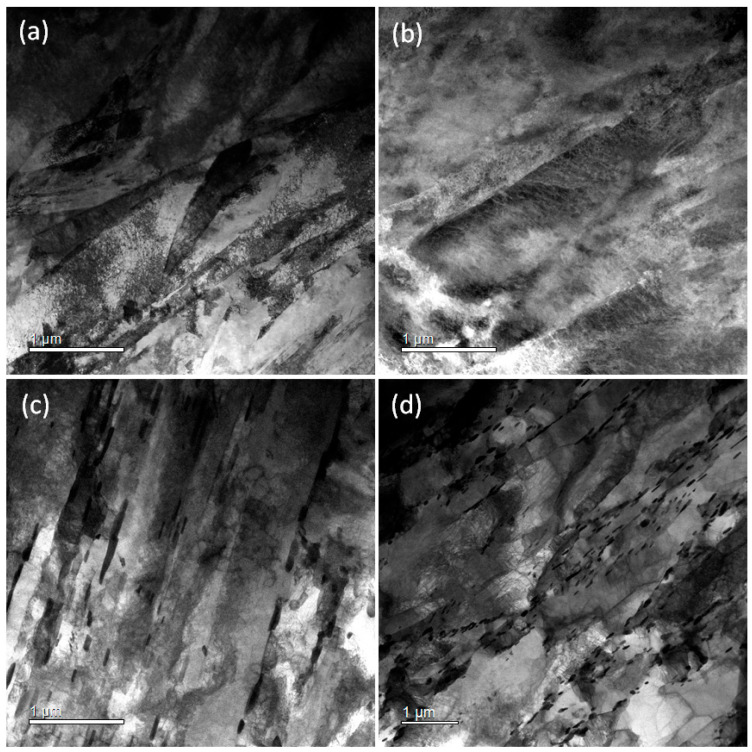
Bright-field TEM images of HT9 steels with (**a**) original composition (heat-3) in as-quenched condition, (**b**) N-addition (heat-4) in as-quenched condition, (**c**) original composition (heat-3) in quenched and 750 °C tempered condition, and (**d**) N-addition (heat-4) in quenched and 750 °C tempered condition. (Note: width of images (**a**) to (**b**) is 3.9 μm, while width of image (**c**) is 6.3 μm).

**Figure 11 materials-17-03803-f011:**
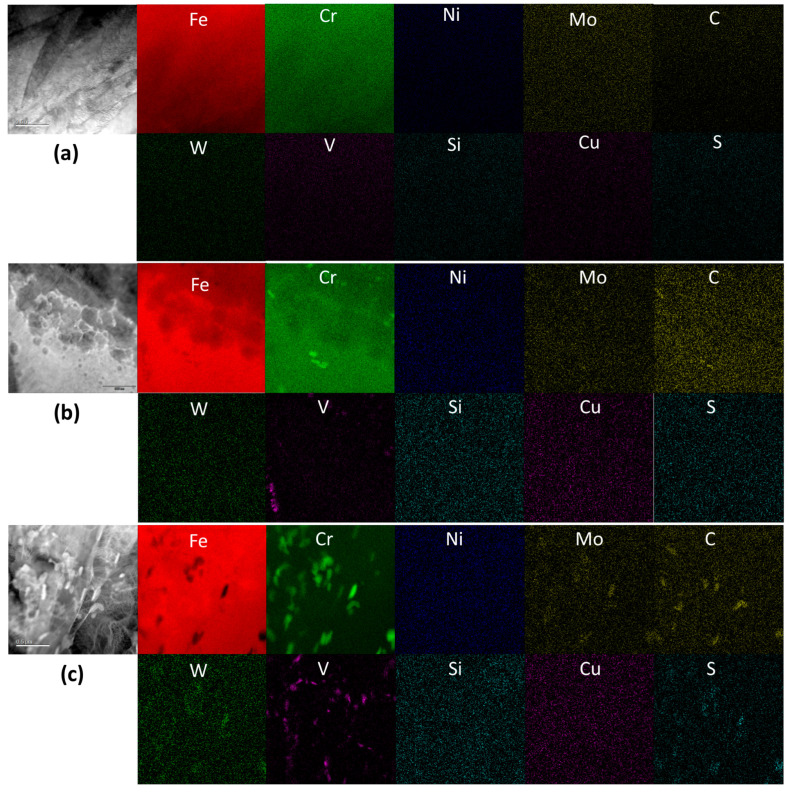
STEM-EDS maps of HT9 steel heat-3 (**a**) in as-quenched condition and (**b**,**c**) in quenched and 750 °C tempered condition. Given in the first column of each group is the corresponding high-angle annular dark-field (HAADF) TEM image. Width of each image corresponds to 2 µm.

**Figure 12 materials-17-03803-f012:**
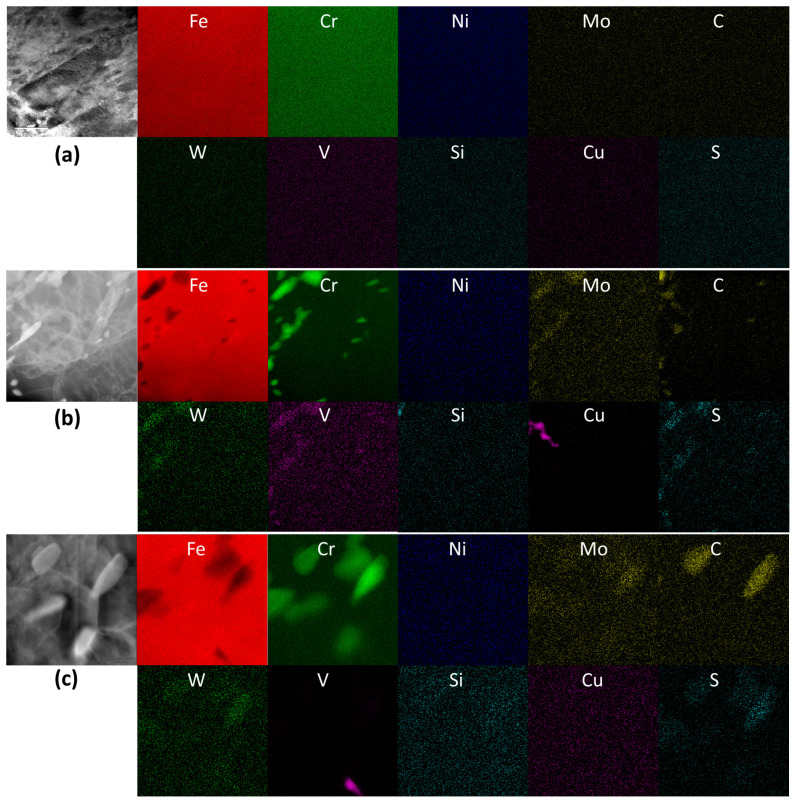
STEM-EDS maps of HT9 steel heat-4 (**a**) in as-quenched condition and (**b**,**c**) in quenched and 750 °C tempered condition. Given in the first column of each group is the corresponding HAADF TEM image. Width of each image corresponds to 4 µm in (**a**), 1 µm in (**b**), and 0.4 µm in (**c**).

**Table 1 materials-17-03803-t001:** Chemical compositions of 12Cr ferritic–martensitic steels with and without nitrogen addition (in wt.%).

Material/Element	Fe	Cr	Mn	Mo	Ni	W	V	Si	C	N
HT9 Steel Heat-3	Bal.	11.07	0.55	1.0	0.51	0.47	0.3	0.25	0.2	0.001
HT9 Steel Heat-4	Bal.	11.42	0.56	1.0	0.52	0.48	0.3	0.26	0.2	0.044

**Table 2 materials-17-03803-t002:** Various thermomechanical processing routes that were used in this research. Tempering processes included both single-step and double-step tempering treatments, after the same normalization and water-quenching treatment.

#	TMP Route	Normalization + Quenching	Tempering-1	Tempering-2
1	#AR (As Rolled)	1100 °C/30 min & Hot Rolling	None	None
2	#WQ (Water Quench)	1070 °C/1 h & WQ	None	None
3	#WQ-300 °C	1070 °C/1 h & WQ	300 °C/1 h & AC	None
4	#WQ-400 °C	1070 °C/1 h & WQ	400 °C/1 h & AC	None
5	#WQ-500 °C	1070 °C/1 h & WQ	500 °C/1 h & AC	None
6	#WQ-600 °C	1070 °C/1 h & WQ	600 °C/1 h & AC	None
7	#WQ-650 °C	1070 °C/1 h & WQ	650 °C/1 h & AC	None
8	#WQ-750 °C	1070 °C/1 h & WQ	750 °C/1 h & AC	None
9	#WQ-300 °C–650 °C	1070 °C/1 h & WQ	300 °C/1 h & AC	650 °C/0.5 h & AC
10	#WQ-400 °C–650 °C	1070 °C/1 h & WQ	400 °C/1 h & AC	650 °C/0.5 h & AC
11	#WQ-500 °C–650 °C	1070 °C/1 h & WQ	500 °C/1 h & AC	650 °C/0.5 h & AC
12	#WQ-600 °C–650 °C	1070 °C/1 h & WQ	600 °C/1 h & AC	650 °C/0.5 h & AC
13	#WQ-300 °C–750 °C	1070 °C/1 h & WQ	300 °C/1 h & AC	750 °C/0.5 h & AC
14	#WQ-400 °C–750 °C	1070 °C/1 h & WQ	400 °C/1 h & AC	750 °C/0.5 h & AC
15	#WQ-500 °C–750 °C	1070 °C/1 h & WQ	500 °C/1 h & AC	750 °C/0.5 h & AC
16	#WQ-500 °C–650 °C-S	1070 °C/1 h & WQ	500 °C/1 h & AC	750 °C/0.25 h & AC
17	#WQ-600 °C–750 °C	1070 °C/1 h & WQ	600 °C/1 h & AC	750 °C/0.5 h & AC
18	#WQ-600 °C–750 °C-S	1070 °C/1 h & WQ	600 °C/1 h & AC	750 °C/0.25 h & AC

Note, # in front of TMP routes indicates heat number (i.e., numbers 3 and 4 indicate Heat-3 and Heat-4, respectively).

## Data Availability

This manuscript has been authored by UT-Battelle, LLC, under contract with the US Department of Energy (DOE). The US government retains and the publisher, by accepting the article for publication, acknowledges that the US government retains a nonexclusive, paid-up, irrevocable, worldwide license to publish or reproduce the published form of this manuscript, or allow others to do so, for US government purposes. DOE will provide public access to these results of federally sponsored research on 31 December 2024, in accordance with the DOE Public Access Plan (http://energy.gov/downloads/doe-public-access-plan, accessed on 28 July 2024).

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
