# Peer review of "Thermomechanical Processing for Improved Mechanical Properties of HT9 Steels"

_materials, 2024, doi:10.3390/ma17153803_

Round 1

Reviewer 1 Report

Comments and Suggestions for Authors

 The investigation provides experimental investigation for steels used at high temperatures. It well written.

1- can the authors provide a further description about the experimental procedure that they used.

2- Can the steels developed in the current investigation be used for temperatures higher than 600 C?  

Author Response

(Reviewer's leading statement) The investigation provides experimental investigation for steels used at high temperatures. It well written.

Response: We appreciate the positive feedback.

Comments 1: Can the authors provide a further description about the experimental procedure that they used?

Response 1: Additional descriptions of specimen orientations and loading directions for tensile and fracture tests have been added on pages 6 and 7 (line 195-202; line 231-234). We believe that all experimental elements necessary for replication by readers are included in the revision, except for the J-R curve construction process. This process is too lengthy to describe in this publication, so we have cited the ASTM standard testing method, which provides a detailed procedure.

Comments 2: Can the steels developed in the current investigation be used for temperatures higher than 600 °C?

Response 2: The authors believe that the maximum application temperature for these materials is approximately 650°C. At around this temperature, the strengthening effects of the non-traditional thermal-mechanical processes (TMPs) become minimal compared to those of materials processed using traditional TMPs. In other words, at approximately 650°C, the influence of different carbide distributions and lath structures is considered negligible, indicating that manipulating TMP has limited effects above this temperature.

Reviewer 2 Report

Comments and Suggestions for Authors

In this manuscript, Thak Sang Byum, et al studied the effects of thermomechanical processing conditions on the microstructural evolution and mechanical performance of ferritic-martensitic steels. It was demonstrated that choosing an appropriate TMP route, including normalization, rapid quenching in water, single- or two-step tempering treatments, significantly influences the mechanical performance of the materials, such as yield stress, ultimate tensile strength, elongation, and fracture toughness. Overall, this is a systematic research effort with a thorough experimental study, offering valuable insights into optimizing thermomechanical processing conditions to achieve both strong and tough ferritic-martensitic steels for developing high-performance core structures in advanced fast reactors. The presented results are interesting and reasonable; therefore, I suggest minor revision with the following suggestions.:

1)     In the current experiments, how many tests were conducted to determine the yield stress and other mechanical properties? Is there a standard deviation for the measured mechanical performance of the materials? The author should provide the details of the experiment.

2)     I recommend that the author include representative stress-strain curves for various materials, as this will help readers more easily differentiate the mechanical performance of the different samples.

3)     In Figures 3 to 6, straight lines are used to connect data points to show trends of the mechanical properties. However, in Figures 7 and 8, curve fitting is employed to illustrate the relationship between the fracture toughness of different samples and the testing temperature. The author should provide an explanation for this difference.

Author Response

(Reviewer’s leading statement): In this manuscript, Thak Sang Byun, et al studied the effects of thermomechanical processing conditions on the microstructural evolution and mechanical performance of ferritic-martensitic steels. It was demonstrated that choosing an appropriate TMP route, including normalization, rapid quenching in water, single- or two-step tempering treatments, significantly influences the mechanical performance of the materials, such as yield stress, ultimate tensile strength, elongation, and fracture toughness. Overall, this is a systematic research effort with a thorough experimental study, offering valuable insights into optimizing thermomechanical processing conditions to achieve both strong and tough ferritic-martensitic steels for developing high-performance core structures in advanced fast reactors. The presented results are interesting and reasonable; therefore, I suggest minor revision with the following suggestions.

Response: We are greatly encouraged by the reviewer’s highly positive assessment.

Comment 1: In the current experiments, how many tests were conducted to determine the yield stress and other mechanical properties? Is there a standard deviation for the measured mechanical performance of the materials? The author should provide the details of the experiment.

Response 1: We conducted only one test per alloy, processing condition, and test condition. As a result, we lack statistical data for each individual data point. The number of tests was too extensive to perform multiple tests under each condition. Consequently, we opted to test each material under various temperatures to identify temperature dependence trends and detect potential outliers. We have included a statement to clarify that each condition was tested only once. This is stated in the beginning of the section 2.3 (Page 6, line 195-202)

Comment 2: I recommend that the author include representative stress-strain curves for various materials, as this will help readers more easily differentiate the mechanical performance of the different samples.

Response 2: Figures 3(a) and 3(b) have been added to the section 3.1 of the manuscript. These figures display two sets of engineering stress-strain curves for the lowest and highest test temperatures (RT and 600°C), respectively. The curves were selected to illustrate the stress-strain behaviors under key TMP conditions: the water-quenched (WQ) condition represents the extremely hardened state of the HT9 steels, while the WQ-750°C condition serves as the reference state. Curves for WQ-650°C and WQ-600°C-650°C are also presented as potentially optimized conditions. A description of the key characteristics of these stress-strain curves has also been added to Section 3.1. (Added line 256 – 288)

Comment 3: In Figures 3 to 6 (now 4 to 7), straight lines are used to connect data points to show trends of the mechanical properties. However, in Figures 7 and 8 (now 8 and 9), curve fitting is employed to illustrate the relationship between the fracture toughness of different samples and the testing temperature. The author should provide an explanation for this difference.

Response 3: The authors appreciate the reviewer pointing this out. These tensile and fracture data come from two different reports that used different styles of graphing. We missed noticing this difference. Figures 4 to 7 (Figures 3 to 6 in original version) have been updated with smooth (fitted) curves. This modification did not introduce any changes to the technical observations.

Reviewer 3 Report

Comments and Suggestions for Authors

The author should consider the following points.

1.  It is interesting to observe that M23C6 precipitates have dissolved before reaching 1050 °C. How is it comparable to earlier works on the same kind of steels? Are the authors confident about the calculations?

2.      Figure 1 illustrates the formation of the laves phase, which occurs at service temperatures. What impact does the laves phase have on the mechanical properties?

3.    The author’s need to establish a connection between microstructural changes resulting from various thermomechanical processes. How do factors such as lath structure, precipitate shape and size, and high/low angle grain boundaries influence the fracture toughness and tensile properties of HT9 steel?

4.      What is lath size in figure 9.

5.      The scale bar in figure 9 (a) and (c) is not clear.

6.      Are the delineated boundaries in figure 9 laths?

7.      From an irradiation standpoint, how does the current study prove useful?

8.   What are the anticipated reasons for the decrease in mechanical strength observed in HT9 steel with nitrogen addition across various TMT conditions?

9.      Please include an image of the experimental setup.

10.  A large number of experimental data on tensile properties and fracture toughness have been reported at different test temperatures and TMP conditions. However, there is no scientific reasoning provided regarding the percentage improvement in mechanical properties with respect to the percentage improvement in lath width and the precipitates (MX/M23C6). To support this, the author may also report SEM images showing the precipitate size distribution in various test conditions."

11.  Author is required to include plot of engineering stress-strain for each condition at different test temperatures.

12.  Why did the newly tested TMPs on heat-4 show minimal improvement in fracture toughness?

Author Response

Comment 1: It is interesting to observe that M23C6 precipitates have dissolved before reaching 1050 °C. How is it comparable to earlier works on the same kind of steels? Are the authors confident about the calculations?

Response 1: The calculation assumed thermodynamic equilibrium, predicting that M23C6 carbides dissolve in the range of 800–1050 °C. In reality, heating and holding the steel at the normalization temperature for an hour may not allow the steel to reach complete thermodynamic equilibrium. Additionally, the earlier hot rolling treatment at 1100 °C might have left some metastable structures, such as dislocations, which could accelerate carbide dissolution. Therefore, we believe the calculation provides a good guide for normalization treatment at 1070 °C. Many other high-chromium steels exhibit similar behavior, with some showing complete carbide dissolution during annealing at 900 °C, depending on their specific alloy composition and microstructure.

Comment 2. Figure 1 illustrates the formation of the laves phase, which occurs at service temperatures. What impact does the laves phase have on the mechanical properties?

Response 2: HT9 steels are designed for use in reactor core components, such as fuel cladding and duct in sodium-cooled reactors. The phase stability under high-temperature and neutron irradiation conditions differs from that under high temperature alone. In particular, alpha-prime and G-phase precipitation might be more relevant to the service conditions, whereas the Laves phase may not appear under high-temperature irradiation conditions. Additionally, the alpha-prime and G-phase formation, as well as carbide dissolution and coarsening, could have detrimental effects on the mechanical properties.

Comment 3: The author’s need to establish a connection between microstructural changes resulting from various thermomechanical processes. How do factors such as lath structure, precipitate shape and size, and high/low angle grain boundaries influence the fracture toughness and tensile properties of HT9 steel?

Response 3: We appreciate this comment/recommendation: the study should include more complete understanding of the roles of the thermomechanical treatments, the under-tempering in particular, and the resultant microstructures being correlated to the mechanical properties. Unfortunately, the microscopy we performed was very limited for pursuing understanding of microstructures and their correlations with the mechanical properties.  Only part of such efforts could be possible in the research.  In the discussion section 4, some discussion on the lath sizes was added (line 519 - 535).  

Comment 4: What is lath size in figure 9 (now figure 10).

Response 4: The lath thicknesses, measured as the intercept lengths, were approximately 280 nm and 470 nm for heat-3 and heat-4 in the as-quenched condition, as shown in Figures 10(a) and 10(b). After tempering, these measurements decreased slightly to about 250 nm and 460 nm, respectively, because the tempering treatment made the lath boundaries more distinct, as illustrated in Figures 10(c) and 10(d). We inserted these sentences in page 19 (also see Response 3, line 519 - 535).

Comment 5: The scale bar in figure 9 (now figure 10) (a) and (c) is not clear.

Response 5: To clarify the scales, a note is added to the Figure 10 caption: Note that the width of images (a) to (b) is 3.9 μm, while the width of image (c) is 6.3 μm.

Comment 6: Are the delineated boundaries in figure 9 laths?

Response 6: We think so. Some lath boundaries in the heat-4 changed to substructure boundaries (laths were divided or developed). Understanding of the detailed character of these boundaries after tempering requires more in-depth study.

Comment 7: From an irradiation standpoint, how does the current study prove useful?

Response 7: Many of these conditions, when under-tempered, result in finer and higher-strength structures. In irradiation scenarios, finer microstructural features can provide more defect-sink sites, potentially delaying the accumulation of radiation damage such as swelling and embrittlement. This research aims to identify a processing route to produce finer structures through rapid quenching and under-tempering, without compromising mechanical properties such as strength, ductility, and fracture toughness. In this context, the research is considered successful, though not substantial.

Comment 8: What are the anticipated reasons for the decrease in mechanical strength observed in HT9 steel with nitrogen addition across various Metaconditions?

Response 8: We anticipated that the nitrogen addition would create more nanoscale hardening particles (nitrides). However, the effect of austenite stabilization predominated in our TMT schedules, which, we believe, reduced the alloy's hardenability and diminished the effectiveness of the nitrogen addition. (Please also refer to Response 12.)

Comment 9. Please include an image of the experimental setup.

Comment 10. A large number of experimental data on tensile properties and fracture toughness have been reported at different test temperatures and TMP conditions. However, there is no scientific reasoning provided regarding the percentage improvement in mechanical properties with respect to the percentage improvement in lath width and the precipitates (MX/MC). To support this, the author may also report SEM images showing the precipitate size distribution in various test conditions.

Response 10: We agree with this recommendation. Microscopy analysis of these newly TMTed materials is limited, although a detailed analysis is required to understand the microstructural evolution during the non-traditional TMT processes and its correlation with the changes of mechanical properties. Unfortunately, the resources within the project were limited, so we would seek another opportunity for a more detailed study.

Comment 11. Author is required to include plot of engineering stress-strain for each condition at different test temperatures.

Response 11: Sixteen (16) selected engineering stress-strain curves were added as in Figures 3(a) and 3(b) along with some description in section 3.1. For the lowest and highest test temperatures (RT and 600°C), eight curves per each temperature were selected for key TMP conditions: the water-quenched (WQ) condition, WQ-750°C condition, and WQ-650°C and WQ-600°C-650°C conditions.  are also presented as potentially optimized conditions. (The same in Response 2 for the reviewer 2).

Comment 12. Why did the newly tested TMPs on heat-4 show minimal improvement in fracture toughness?

Response 12: The main merits of ferritic-martensitic steels, such as HT9 steel, include their high hardenability (ability to freeze to a fine lath structure), which can lead to fine microstructures and, consequently, a good combination of strength and toughness (ductility). The nitrogen addition in Heat 4 was intended to produce nitride particles, which may have higher thermal stability than carbides. However, this study indicates that nitrogen addition reduces the hardenability of the alloy (since nitrogen acts as an austenite stabilizer) and results in a microstructure that is not finer, showing no desired improvement in fracture toughness or strength.